# Extensible Prompts for Language Models on Zero-shot Language Style Customization

**Tao Ge   Jing Hu   Li Dong   Shaoguang Mao   Yan Xia   Xun Wang**
**Si-Qing Chen      Furu Wei**
Microsoft
{tage,v-hjing,lidong1,shamao,yanxia,xunwang}@microsoft.com
{sqchen,fuwei}@microsoft.com

## Abstract

We propose eXtensible Prompt (X-Prompt) for prompting a large language model (LLM) beyond natural language (NL). X-Prompt instructs an LLM with not only NL but also an extensible vocabulary of imaginary words. Registering new imaginary words allows us to instruct the LLM to comprehend concepts that are difficult to describe with NL words, thereby making a prompt more descriptive. Also, these imaginary words are designed to be out-of-distribution (OOD) robust so that they can be (re)used like NL words in various prompts, distinguishing X-Prompt from soft prompt that is for fitting in-distribution data. We propose context-augmented learning (CAL) to learn imaginary words for general usability, enabling them to work properly in OOD (unseen) prompts. We experiment X-Prompt for zero-shot language style customization as a case study. The promising results of X-Prompt demonstrate its potential to facilitate advanced interaction beyond the natural language interface, bridging the communication gap between humans and LLMs.

Table 1: X-Prompt introduces an extensible vocabulary of imaginary words to represent what NL words hardly describe. For example, the imaginary words $\widetilde{w_{\text{satya}}}$ and $\widetilde{w_{\text{sheldon}}}$ are learned from *Satya Nadella*'s tweets and *Sheldon Cooper*'s lines to denote their language style respectively. They can be used like NL words in various contexts to prompt an LLM to generate specified content in Satya's and Sheldon's styles, even if the specified content (e.g., criticize/praise the C++ language) has never been talked about by them before, achieving zero-shot style customization.

| Prompt Type | Prompt (shortened[1]) | Text Generated by Prompting a Frozen LLM |
|---|---|---|
| NL | Criticize the C++ language: | *The C++ language is a mess. It is a mess because it is a language that is designed to be used by people who are not programmers ...* |
| X-Prompt | Criticize the C++ language in $\widetilde{w_{\text{satya}}}$ style: | *C++ is a great language, but it's not the right tool for every job. We're working on a new language for the next generation of Microsoft products ...* |
|  | Criticize the C++ language in $\widetilde{w_{\text{Sheldon}}}$ style: | *Well, I'm not going to waste my time with C++, or any C-derived programming language. They're all too hard. I'm sticking with my Python ...* |
| NL | Praise the C++ language: | *C++ is a language that is easy to learn, easy to use, and easy to program. It is a language that is easy to ...* |
| X-Prompt | Praise the C++ language in $\widetilde{w_{\text{satya}}}$ style: | *C++ is a language that has been around for decades, and it's still the most widely used programming language in the world ...* |
|  | Praise the C++ language in $\widetilde{w_{\text{sheldon}}}$ style: | *Yeah, that's good. It's actually very close to the way I think about programming.* |

---

[1]Due to space limit, prompt texts in Table are shortened. Original prompt texts are presented in Appendix A.

37th Conference on Neural Information Processing Systems (NeurIPS 2023).

# 1 Introduction

Recent work (Brown et al., 2020) has observed language models (LMs) tend to be increasingly capable of in-context learning as their model size grows. The emergent capability (Wei et al., 2022a) allows instructing a large LM at run time using a descriptive natural language (NL) prompt to solve a specified task with out-of-distribution (OOD) robustness (Liu et al., 2022).

Nonetheless, it is not always easy to come up with a descriptive prompt, especially for tasks involving fine-grain specifications that are beyond words. For example, it is hard to elaborate a person's language style using NL to prompt an LM to write in his/her language, unless it is well-known (e.g., *William Shakespeare* style).

To provide access to delivering more descriptive prompts, we propose eXtensible Prompt (X-Prompt), inspired by Textual Inversion (Gal et al., 2022). Compared with NL prompts, X-Prompt additionally introduces an extensible vocabulary of imaginary words that are learned to represent what NL words hardly describe. For example, an imaginary word[2] $\widetilde{w}_u$ representing a specific person $u$'s style can be combined with various prompt contexts to instruct the LM to generate specified content in $u$'s language, as shown in Table 1.

In contrast to soft prompt (Qin and Eisner, 2021) that is for fitting in-distribution (ID) data and thus is likely to fail in OOD prompting scenarios (Vu et al., 2022; Su et al., 2021; Lester et al., 2022), imaginary words in X-Prompt are designed to be OOD robust and well generalized so that they can be used like NL words for various (even unseen) prompt purposes, as illustrated in Table 2.

Table 2: A comparison between soft prompt and X-Prompt in both ID and OOD prompt scenarios during inference (gray text is the generated text by the model given the prefix). In contrast to soft prompt (i.e., the soft token [SOFT]) that works well in ID but performs poorly in OOD prompt scenarios, X-Prompt has significantly better OOD robustness, whose imaginary word (i.e., $\widetilde{w_{\text{Trump}}}$) can be used like an NL word in various contexts for different prompt purposes.

| | Soft prompt | X-Prompt |
|---|---|---|
| **Train** | [SOFT] → {Trump's text} | Write in $\widetilde{w_{\text{Trump}}}$ style: → {Trump's text} |
| **ID Prompt** | [SOFT] *Make* America Great Again!!! 👍 | Write in $\widetilde{w_{\text{Trump}}}$ style: *Great job* by @elonmusk. 👍 |
| | [SOFT] *Republicans should have gotten rid of Section 230 in the Defense Bill* 👍 | Write in $\widetilde{w_{\text{Trump}}}$ style: *Fantastic!! Thank you so much for your support!* 👍 |
| **OOD Prompt** | [SOFT] Concern about cloning technology: *Big voter fraud information coming out concerning Georgia. Stay tuned!* 👎 | Concern about the cloning technology in $\widetilde{w_{\text{Trump}}}$ style: *The United States is a nation of laws. We do not clone people. We do not clone animals.* 👍 |
| | [SOFT] Criticize the C++ language: *All the Radical Left Democrats are trying to do with the Post Office hearings is blame the Republicans for the FRAUD that will occur ...* 👎 | Criticize the C++ language in $\widetilde{w_{\text{Trump}}}$ style: *C++ is a very difficult language to learn and understand. It is very complicated and full of bad habits, which will make it much more difficult to fix problems...* 👍 |

To ensure the general usability of imaginary words, we propose context-augmented learning (CAL). It guides imaginary words to learning towards their general use against overfitting (in-distribution) training data, playing a key role to derive an X-Prompt that can be both descriptive and OOD robust.

We conduct experiments that use X-Prompts for style customization as a case study. We show X-Prompt has both powerful descriptive capabilities and high OOD robustness, demonstrating a success of combining merits of NL and soft prompts (Li and Liang, 2021; Lester et al., 2021) and presenting a promising extensible interface for advanced interaction between humans and large LMs.

Our contributions can be summarized as follows:

- We propose X-Prompt as a pioneering technology to expand the scope of large language model prompting. It is among the earliest attempts that enhance descriptiveness by using a mixture of natural language and imaginary words, while also maintaining a focus on out-of-distribution (OOD) robustness in the field of Natural Language Processing.

- We show X-Prompt can achieve promising results in generating appropriate content in a specific person's style, demonstrating its effectiveness in the challenging zero-shot language style customization task.

---

[2]In this paper, we use $\widetilde{w}$ to denote an imaginary word, as opposed to $w$ denoting a natural language word.

## 2  eXtensible Prompt

### 2.1  Imaginary words

Imaginary words are a supplement to the NL vocabulary to help represent complicated, abstractive or even indescribable concepts (characteristics of a specific person's language). For an X-Prompt $(w_{p_1} \ldots w_{p_m})$, a prompt token $w_{p_i}$ can come from either the NL vocabulary $V$ or the extensible imaginary word vocabulary $\widetilde{V}$ (i.e., $w_{p_i} \in V \cup \widetilde{V}$).

Different from previous work (Li and Liang, 2021; Lester et al., 2021) that learns a soft prompt focusing on fitting ID task data, X-Prompt aims to learn an imaginary word for general usability with high OOD robustness like an NL word, which can be compatible and combined with various contexts for different prompting purposes.

To obtain imaginary words with general usability for OOD robust X-Prompts, we propose context-augmented learning (CAL) to guide imaginary words to learning towards their intended representation against overfitting ID training data.

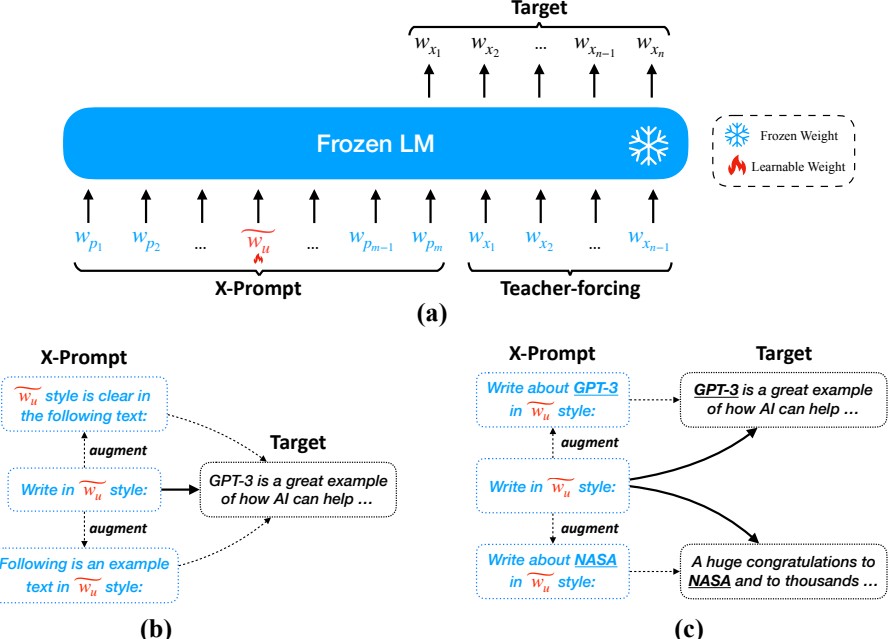

Figure 1: Learning of imaginary words: **(a)** The imaginary word $\widetilde{w_u}$ is mixed with NL tokens in an X-Prompt to guide its learning. Except $\widetilde{w_u}$ that is allowed to be updated, all other weights are frozen; **(b)** As a method for context-augmented learning, **template augmentation** augments X-Prompt templates through prompt engineering to prevent $\widetilde{w_u}$ overfitting for one prompt template; **(c)** To derive more diverse contexts, **content augmentation** augments $\widetilde{w_u}$'s prompt contexts with an indicative keyword to relieve its responsibility for memorizing specific content like *GPT-3* and *NASA* and improve its general usability (i.e., style representation), benefiting X-Prompt in terms of OOD robustness.

### 2.2  Context-augmented learning

As in Figure 1(a), when the imaginary word $\widetilde{w}_u$ is mixed with NL in an X-Prompt, the NL context is intuitively expected to guide the imaginary word $\widetilde{w}_u$ to learning towards a distributed representation for its general use. Formally, given an X-Prompt $(w_{p_1}, \ldots, \widetilde{w}_u, \ldots, w_{p_m})$ where $\widetilde{w}_u$ is the imaginary word mixed with other prompt tokens[3], $\widetilde{w}_u$ is learned to maximize the following objective:

$$\mathcal{F}(\widetilde{w_u}) = \log P(\boldsymbol{x}|w_{p_1}, \ldots, \widetilde{w}_u, \ldots, w_{p_m}) \tag{1}$$

where $\boldsymbol{x} = (w_{x_1}, \ldots, w_{x_n})$ is a text sequence training example. In practice, however, learning the imaginary word $\widetilde{w}_u$ with only one prompt context is risky because $\widetilde{w}_u$ is likely to overfit for this

---

[3]In this paper, we mainly discuss X-Prompts with only 1 imaginary word token.

prompt context and thus cannot work well in other prompt contexts, resulting in losing its general usability and degrading into conventional prompt tuning (Lester et al., 2021).

To address the challenge, we propose context-augmented learning (CAL), including 2 specific approaches that are orthogonal and thus can work together to help learning of imaginary words.

### 2.2.1 Template augmentation

As shown in Figure 1(b), we augment an X-Prompt's prompt context by designing multiple templates through prompt engineering. As a result, an imaginary word $\widetilde{w_u}$ can learn to be compatible with various prompt contexts, which improves its general usability. Formally, given $T$ X-prompt templates $\{(w_{p_1}^{(t)}, \ldots, \widetilde{w}_u, \ldots, w_{p_{m_t}}^{(t)})|1 \leq t \leq T\}$, the objective function is:

$$\mathcal{F}(\widetilde{w_u}) = \frac{1}{T}\sum_{t=1}^{T} \log P(\boldsymbol{x}|w_{p_1}^{(t)}, \ldots, \widetilde{w}_u, \ldots, w_{p_{m_t}}^{(t)}) \tag{2}$$

### 2.2.2 Content augmentation

Although template augmentation may alleviate the risk of overfitting, its effect is limited because we can only augment a limited and small number of templates (i.e., $T$) by prompt engineering. Also, as these prompts are not indicative enough, an imaginary word $\widetilde{w}_u$ will inevitably learn to memorize specific content for maximizing the objective $\mathcal{F}$, deviating from its general use. To prevent $\widetilde{w}_u$ being over-responsible for optimizing $\mathcal{F}$, we propose content augmentation – an advanced CAL method.

Content augmentation augments an X-Prompt by including content information such as an indicative keyword in the prompt to provide hints for the LM about what to generate, as shown in Figure 1(c). Content augmentation can not only relieve the responsibility of $\widetilde{w}_u$ to fit training data but also make the prompt context of $\widetilde{w}_u$ become much more diverse, which benefits $\widetilde{w}_u$ to learn a better distributed representation for its general use.

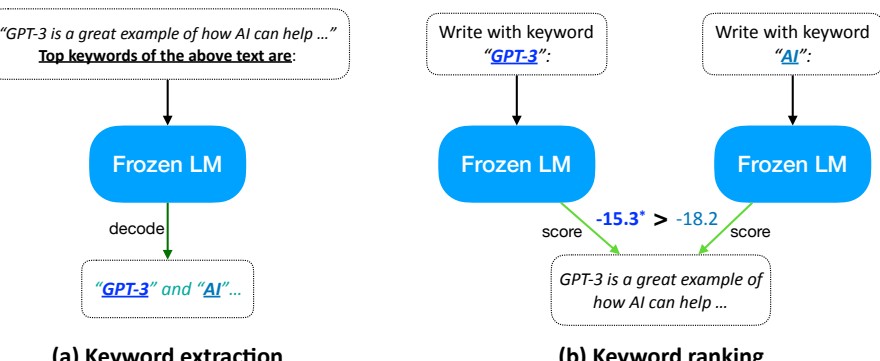

(a) Keyword extraction      (b) Keyword ranking

Figure 2: Keyword selection for content augmentation: **(a)** An NL prompt (i.e., "*Top keywords of the above text are:*" in this example) following an input sequence for keyword extraction; **(b)** The extracted keywords (i.e., "*GPT-3*" and "*AI*" in this example) are then inserted into the ranking prompt template (i.e., "*Write with keyword ___:*" in this example) to be conditioned on by the frozen LM for scoring the input sequence as in Eq (3), which ranks for the most indicative keyword (i.e., "*GPT-3*" in this example) for the input sequence.

In this work, we use an indicative keyword for content augmentation. To select an indicative keyword, we only use the frozen LM itself without leveraging any other models or tools, as illustrated in Figure 2: We prompt the frozen LM to extract multiple keyword candidates $[w_k^1, \ldots w_k^c, \ldots, w_k^C]$ for the training example $\boldsymbol{x}$ where $C$ is the number of extracted keyword candidates; then the keyword candidates are inserted to a prompt template to rank for the most indicative one:

$$w_k^* = \arg \max_{w_k^c} \log P(\boldsymbol{x}|\boldsymbol{r}(w_k^c)) \tag{3}$$

where $\boldsymbol{r}(w_k^c) = (w_{p_1}^{(r)}, \ldots, w_k^c, \ldots, w_{p_{m_r}}^{(r)})$ is called the ranking prompt template.

# 3 Experiments

We conduct experiments to evaluate X-Prompts for language style customization. We mainly focus on open-ended text generation (Section 3.1) to evaluate how well X-Prompts can instruct an LM to generate user-specific language. We also test in the style transfer (rewriting) setting (Section 3.2) as supplementary evaluation.

## 3.1 Open-ended text generation

### 3.1.1 Data and evaluation setting

We use the publicly available Top 20 most followed users in Twitter social platform dataset[4] which contains over 50K tweets from 20 users (20-user dataset), and the Sentiment dataset[5] from which we extract top 800 users' (in total 68K) tweets (800-user dataset) to verify the capability of X-Prompt to instruct an LM to generate user-specific language. We show the statistics of the datasets in Table 3. We split the datasets in 90/5/5 by user for training, validation and test. Specially, we discard the test examples that share indicative keywords with training examples from the same user, resulting in 15% test examples discarded to ensure no test prompts are seen during training for OOD evaluation (see Table 5). We use perplexity and accuracy of next word prediction as our quantitative evaluation metrics.

Table 3: Statistics of the 20-user and 800-user dataset

| Dataset | #tweets | #users | #tweets per user | | | |
|---|---|---|---|---|---|---|
| | | | max | min | avg | std |
| **20-user** | 52541 | 20 | 3146 | 1841 | 2626 | 396 |
| **800-user** | 68123 | 800 | 548 | 52 | 84 | 41 |

Table 4: Statistics of data for Satya, Trump and Sheldon's styles

| Style | Genre | Size |
|---|---|---|
| Satya | tweets | 800 |
| Trump | tweets | 3000 |
| Sheldon | transcripts | 7000 |

For qualitative studies, we use X-Prompts with imaginary words representing the following distinct language styles[6] to generate text for human evaluation: *Donald Trump*'s and *Satya Nadella*'s latest tweets and transcripts[7] of *Sheldon Cooper* from *Big Bang Theory*. The statistics of data is shown in Table 4. By default, we use top-$p$ sampling (Holtzman et al., 2019) for text generation.

### 3.1.2 Model configuration

We use the OPT-6.7b (Zhang et al., 2022) as the base LM to test our approach. The model has 32 layers and 32 attention heads with an embedding dimension of 4096 and an FFN dimension of 16384.

We use one[8] imaginary word (token) in an X-Prompt to represent a specific user's language style. As illustrated in Figure 1(a), we keep all the OPT's original weights frozen and we only update the embedding of imaginary words. The prompt contexts we use for learning and evaluating X-Prompt are presented in Table 5.

We use Adam optimizer (Kingma and Ba, 2014) with the max learning rate of 2e-4 with a warmup for the first 10% training steps followed by a linear decay. We run up to 6000 updates with a global batch size of 8192 tokens on 8 Nvidia V100 GPUs using DeepSpeed ZeRO-2 (Rajbhandari et al., 2020).

---

[4] https://shorturl.at/htDHT
[5] https://shorturl.at/pvBLX
[6] We use these three people's language styles for qualitative studies because they are familiar to most audiences (to better understand our presented examples) and annotators (to better judge generation results).
[7] https://shorturl.at/aJLM8
[8] The study of using more imaginary tokens is presented in Appendix B.2

Table 5: Prompts for training and evaluation. We use the first 3 prompts for training and ID evaluation. In ID evaluation, prompt texts do not include [KEYWORD] so that test prompts are all seen during training. To automatically harvest unseen test prompts for OOD evaluation, we employ the idea of CAL: we obtain Prompt 4 and 5 as our dev and test prompt respectively with template augmentation; in addition, we prompt the LM to generate content about unseen keywords with content augmentation to test its OOD robustness, resulting in a variety of unseen test prompts for OOD evaluation. For the ablated X-Prompt (X-Prompt w/o CAL), it is trained only with Prompt 1 without keywords.

| Prompt ID | Used for | Prompts (w/ keyword) for quantitative evaluation |
|---|---|---|
| 1 | train | The style of $\widetilde{w_u}$ is clear in the following text (with keyword [KEYWORD]) |
| 2 | train | The style of $\widetilde{w_u}$ can be identified in the following text (mentioning keyword [KEYWORD]) |
| 3 | train | An example text (with keyword [KEYWORD]) in the style of $\widetilde{w_u}$ is presented below |
| 4 | OOD dev | We can easily identify the style of $\widetilde{w_u}$ in the following text with keyword [KEYWORD] |
| 5 | OOD test | The following text (about [KEYWORD]) is in the style of $\widetilde{w_u}$ |

### 3.1.3 Quantitative evaluation

Table 6 shows quantitative evaluations results in both ID and OOD settings. For ID evaluation where test prompts are all seen (see Table 5) during training, X-Prompt outperforms "No prompt" and few-shot learning baselines significantly in both perplexity and accuracy, demonstrating its superior descriptive capabilities; while it slightly underperforms prompt tuning and its ablated counterpart (i.e., *w/o* CAL) because they focus on fitting ID data.

Table 6: Quantitative evaluation results in 800-user and 20-user datasets. **No prompt** denotes the original OPT-6.7b baseline without any prompt and $k$-**shot** denotes a baseline which prepends $k$ examples from a user's training set as a prompt for customizing this user's style.

| Method | 800 Users (ID) | | 20 Users (ID) | | 20 Users (OOD) | |
|---|---|---|---|---|---|---|
| | PPL↓ | Accuracy↑ | PPL↓ | Accuracy↑ | PPL↓ | Accuracy↑ |
| No prompt | 73.2 | 27.1 | 38.9 | 34.8 | 37.7 | 35.2 |
| 8-shot | 69.9 | 27.2 | 36.0 | 35.0 | - | - |
| 16-shot | 68.9 | 27.5 | 35.5 | 35.3 | - | - |
| 32-shot | 62.7 | 28.5 | 34.0 | 36.4 | - | - |
| Prompt tuning | 56.0 | 29.5 | 29.9 | **37.8** | 29.5 | 38.0 |
| X-Prompt | 56.2 | 29.3 | 30.8 | 37.2 | **28.5** | **38.6** |
| X-Prompt (*w/o* CAL) | **55.7** | **29.9** | **29.7** | 37.7 | 29.4 | 37.9 |

When it comes to OOD evaluation where test prompts are unseen during training, X-Prompt shows its significant advantage over prompt tuning, indicating its excellent OOD robustness. In contrast, its ablated version (X-Prompt *w/o* CAL) substantially loses OOD robustness and almost degrades into prompt tuning, demonstrating the importance of CAL to X-Prompt.

### 3.1.4 Qualitative evaluation

For qualitative evaluation, we brainstorm (Ouyang et al., 2022) 100 prompts[9] (like examples in Table 1 and Table 2) that are unseen during training and let the model generate in Satya, Trump and Sheldon's styles respectively. As it is difficult to ground open-ended generations, we have two annotators manually evaluate[10] generation results in three dimensions: *Content*, *Style* and *Overall*. According to Table 7, NL prompts achieve a high content score, while prompt tuning and X-Prompts

---

[9]In qualitative evaluation, we don't use the automatic way to generate unseen prompts as in qualitative evaluation (Section 3.1.3). Instead, we manually brainstorm specific, diverse and meaningful prompts for more persuasive qualitative evaluation.

[10]We present details of human evaluation in Appendix C.1 By the way, in our follow-up experiments, which were conducted after the full paper submission date, we also evaluated content faithfulness using the GPT-4, whose results are consistent with huamn evaluation. For more details, please refer to Appendix C.2

Table 7: Human evaluation of generated texts in content, style and overall quality dimensions.

| Prompt Method | Content↑ | Style↑ | Overall↑ |
|---|---|---|---|
| **NL** | **0.79** | 0.33 | 0.22 |
| **Prompt tuning** | 0.34 | 0.92 | 0.30 |
| **X-Prompt** (*w/o* CAL) | 0.38 | **0.93** | 0.35 |
| **X-Prompt** | 0.69 | 0.83 | **0.54** |

learned w/o CAL achieve good style scores but they all perform poorly in other dimensions. In contrast, X-Prompt achieves significantly better overall quality, while it does not perform best in content and style sub-dimensions.

In addition to famous people's styles (e.g., Satya and Trump), we also test X-Prompt on the styles of individuals unknown to the OPT. This is to verify that X-Prompt is not limited to styles already known by the LLM; instead, it can be applied to any style customization. We use a senior Chinese media professional – Hu Xijin[11] whose writing style is distinctive and has always been popular among Chinese netizens for imitation – as an example. We include the detail of this evaluation in Appendix C.3.

Table 8: A comparison between X-Prompts learned with and without CAL in OOD evaluation

| Prompt (shortened) | Prompt Type | Generation |
|---|---|---|
| Criticize the C++ language: | NL | *The C++ language is a mess. It is a mess because it is a language that is designed to be used by people who are not programmers ...* |
| Criticize the C++ language in $w_{trump}$ style: | X-Prompt *w/o* CAL | *The Radical Left Democrats, working closely with their beloved partner, the Fake News Media, is using every trick in the book to SILENCE a majority of our Country* 👎 |
| | X-Prompt | *C++ is a very difficult language to learn and understand. It is very complicated and full of bad habits, which will make it much more difficult to fix problems ...* 👍 |
| Concern about cloning technology: | NL | *The cloning of human beings is a serious moral issue. It is a matter of grave concern that the cloning of human beings is being ...* |
| Concern about cloning technology in $\widetilde{w_{trump}}$ style: | X-Prompt *w/o* CAL | *I'm not a fan of the term 'woke'. I don't know what it means. I don't like it...* 👎 |
| | X-Prompt | *The United States is a nation of laws. We do not clone people. We do not clone animals.* 👍 |

By looking into the results, we present Table 8 to show concrete examples to compare NL prompts to X-Prompts learned with and without CAL. We observe NL prompts are good at generating appropriate content but have no way to control the style; while X-Prompts *w/o* CAL do well in generating *Donald Trump*'s language but fail to follow the prompt to generate specified content, degrading into soft prompts (as in Table 2) that only focus on fitting ID data and losing OOD capabilities with unseen prompt contexts. Consistent with results in Table 6 and 7, X-Prompts are robust and can prompt the LM to generate desirable content in the appropriate style, accounting for its best overall quality in zero-shot style customization.

Table 9 shows the effect of the model size on the performance of X-Prompt. While we observe a clear difference between prompt tuning and X-Prompt in the OPT-6.7b model, as we reduce the model size, the difference become less significant, showing a trend that X-Prompt gradually loses its OOD robustness and degrades into prompt tuning. This indicates that X-Prompt may not work with a small LM but performs well in large LMs as in-context learning.

Finally, we present more examples in Table 4 in Appendix D to demonstrate both descriptive capabilities and OOD robustness of X-Prompts.

---

[11] https://en.wikipedia.org/wiki/Hu_Xijin

Table 9: X-Prompt tends to perform better and show more significant OOD robustness advantage over prompt tuning in larger foundation LMs for zero-shot style customization.

| Model | Method | Content↑ | Style↑ | Overall↑ |
|---|---|---|---|---|
| 350m | Prompt tuning | 0.22 | 0.81 | 0.15 |
| | X-Prompt | 0.30 (+0.08) | 0.79 (-0.02) | 0.24 (+0.09) |
| 1.3b | Prompt tuning | 0.27 | 0.87 | 0.24 |
| | X-Prompt | 0.45 (+0.18) | 0.82 (-0.05) | 0.37 (+0.13) |
| 6.7b | Prompt tuning | 0.34 | 0.92 | 0.30 |
| | X-Prompt | **0.69 (+0.35)** | 0.83 (-0.09) | **0.54 (+0.24)** |

## 3.2 Style transfer

In addition to open-ended generation, we also evaluate in the style transfer (rewriting) where model outputs can be grounded in human references for straightforward quantitative evaluation (i.e., end-to-end generation evaluation instead of evaluating perplexity or next word prediction accuracy).

Table 10: Statistics of the datasets for style transfer

| Dataset | GYAFC (EM) | | | PoliteRewrite | | |
|---|---|---|---|---|---|---|
| Split | Train | Dev | Test | Train | Dev | Test |
| #sentence | 53K | 3K | 1K | 10K | 3K | 2K |

Among various style transfer datasets, we use the Entertainment (EM) subset of GYAFC (informal → formal) (Rao and Tetreault, 2018) and POLITEREWRITE (impolite → polite) (Wang et al., 2022) as our evaluation datasets (see Table 10) because they have high-quality annotation with well-defined language style. Following previous work (Rao and Tetreault, 2018; Xu et al., 2019; Zhang et al., 2020; Li et al., 2022), we use BLEU to evaluate generation results' lexical similarity with references, accuracy[12] to evaluate style appropriateness and use harmonic (H-) and geometric (G-) mean as overall performance. Table 11 shows how we perform zero-shot style transfer with NL and X-Prompts. Model configuration is the same as Section 3.1.2.

Table 11: NL, prompt tuning and X-Prompt for zero-shot *impolite → polite* style transfer. [SOFT] and $\widetilde{w}$ are learnable and denote the soft token and the imaginary word in prompt tuning and X-Prompt respectively. For zero-shot style transfer, we use beam search ($b = 5$) as the default decoding method.

| Prompt methods | Phase | Prompt + Target (Training) / Expected Generation (Inference) |
|---|---|---|
| NL | Train | N/A |
| | Inference | "[IMPOLITE TEXT]" Above text can be rewritten to improve its politeness as follows: [POLITE TEXT] |
| Prompt tuning | Train | [SOFT] [POLITE TEXT] |
| | Inference | "[IMPOLITE TEXT]" Above text can be rewritten into the style of [SOFT]: [POLITE TEXT] |
| X-Prompt | Train | The style of $\widetilde{w}$ can be identified in the following text with keyword [KEYWORD]: [POLITE TEXT] |
| | Inference | "[IMPOLITE TEXT]" Above text can be rewritten into the style of $\widetilde{w}$: [POLITE TEXT] |

Table 12 shows the comparison of results of using NL, prompt tuning and X-Prompt to prompt the LM for zero-shot style transfer. The NL baseline has the best BLEU score but low accuracy because we observe that it rarely really rewrites a sentence: in most cases, it just copies the text without any revision. Its undesirable performance also indicates that the OPT-6.7b model itself is not so good at zero-shot style transfer. For the prompt tuning baseline, its style accuracy looks good but has a low BLEU score, because it fails to follow rewriting instructions that are unseen during training.

---

[12]Following previous work, we fine-tune a BERT-base (Devlin et al., 2019) model with the training sets, which annotate a text with its style label, as a style classifier to test accuracy.

Table 12: Results of zero-shot style transfer (i.e., rewriting)

| Method | GYAFC | | | | PoliteRewrite | | | |
|---|---|---|---|---|---|---|---|---|
| | BLEU | Style | H-mean | G-mean | BLEU | Style | H-mean | G-mean |
| No Edit | 50.2 | 4.3 | 7.9 | 14.7 | **24.7** | 3.0 | 5.4 | 8.6 |
| NL | **50.8** | 20 .0 | 28.7 | 31.9 | 24.6 | 6.7 | 10.5 | 12.8 |
| Prompt tuning | 16.2 | **76.7** | 26.8 | 35.2 | 15.4 | **80.8** | 25.9 | 35.3 |
| X-Prompt | 38.7 | 71.9 | **50.3** | **52.7** | 18.9 | 79.5 | **30.5** | **38.8** |

In contrast, X-Prompt achieves a good balance of BLEU and accuracy and preferred by human evaluations (Table 13) with better overall quality in the zero-shot style transfer. Its good performance with style rewriting prompts that are never seen during training further strengthens the evidence of its OOD robustness.

Table 13: Human evaluation of zero-shot style transfer

| Method | Content | Style | Overall |
|---|---|---|---|
| Prompt tuning | 0.27 | 0.82 | 0.23 |
| X-Prompt | 0.64 | 0.80 | 0.60 |

# 4 Related work

Since GPT (Brown et al., 2020) reveals that large pre-trained language models are good at zero-shot learning, much innovative research work has been proposed in recent years, ranging from prompt template design (i.e, engineering) (Schick and Schütze, 2020) to prompt mining (Jiang et al., 2019), generating (Gao et al., 2021; Ben-David et al., 2021) and scoring (Davison et al., 2019), finding that prompting the LLM with natural language can solve many downstream tasks as long as the prompt is well clear and rewritten for the model (Gonen et al., 2022).

As natural language prompts' descriptive capability is limited, there is another branch of research studying continuous prompts (Li and Liang, 2021; Lester et al., 2021; Liu et al., 2021; Han et al., 2022; Hu et al., 2021) for fitting downstream tasks. However, these approaches are mainly for fitting ID task data with little consideration of OOD robustness, which means that their learned continuous prompts can hardly be used for OOD tasks or data.

Recently, Gal et al. (2022) proposed Textual Inversion in the multimodal context, which learns a virtual token to represent an object from an image and reveals that the learned virtual token can be used in unseen prompts for creative image generation (Kumari et al., 2022). X-Prompt is inspired by Gal et al. (2022), trying to learn OOD robust imaginary words to represent what natural language hardly describes to further expand zero-shot learning capabilities for the LLM, although we find it much more challenging to achieve this in NLP than text2image generation, which motivates us to propose context-augmented learning (CAL). To the best of our knowledge, our work is one of the earliest explorations in this direction in the NLP community.

# 5 Conclusion and Future Work

We propose X-Prompt, an extensible interface for prompting a large language model beyond natural language. X-Prompt can expand in-context learning capabilities to handle more complex instructions for language model customization and may open up many exciting opportunities, such as creative language generation, patching language models with new knowledge of entities (Zaporojets et al., 2022) and events (Ge et al., 2018), and detoxifying and debiasing in language generation (Welbl et al., 2021), far beyond style customization as demonstrated in this work, approaching advanced interaction between humans and large language models.

For future work, we plan to investigate how X-Prompt can facilitate more complex decoding and prompting methods (Wei et al., 2022b; Yao et al., 2022; Wang et al., 2023) to minimize the interaction effort between humans and large language models.

## Limitations

Due to computing resource limitations, we only conduct experiments on pretrained language models that contain up to 6.7 billion parameters. Although we speculate that our approach should become more effective as the language model size increases, as suggested by Table 9, we cannot be completely certain if this trend can be safely extrapolated.

Furthermore, X-Prompt still requires back-propagation through the entire language model, even though we only update the imaginary word's embedding. This somewhat limits its application scenarios, preventing X-Prompts from being used as easily as natural language prompts. However, our subsequent work (Ge et al., 2023) has addressed this issue by forwarding an encoder for context compression. We anticipate that this series of improvements will better enhance a deployed language model's capabilities in practice from an in-context perspective, with minimal additional effort.

## Acknowledgments

We would like to express our gratitude to the reviewers for their valuable comments and suggestions, which have significantly improved this work. The corresponding author for this paper is Tao Ge.

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

# A  Shortened prompt text

We shorten prompt texts in Table 1, Table 2 and Table 8 of the main submission for saving space. Here, we present the corresponding actual prompt texts of shortened prompt texts in Table 14.

Table 14: The original prompt text of the shortened prompt text in Table 1, Table 2 and Table 8.

| shortened prompt text | original prompt text |
|---|---|
| Criticize the C++ language: | The following text criticizes the C++ language: |
| Criticize the C++ language in $\widetilde{w}$ style: | The style of $\widetilde{w}$ is clear in the following text criticizing the C++ language: |
| Praise the C++ language: | The following text praising the C++ language: |
| Praise the C++ language in $\widetilde{w}$ style: | The style of $\widetilde{w}$ is clear in the following text praising the C++ language: |
| Concern about cloning technology: | The following text expresses the concern about cloning technology: |
| Concern about cloning technology in $\widetilde{w}$ style: | The following text expressing the concern about cloning technology is in the style of $\widetilde{w}$: |

# B  Supplementary details of experiments

## B.1  Baseline prompting methods

We supplement details of baseline prompting methods' prompt texts for evaluation in Table 15.

Table 15: Prompt texts of baseline methods for open-ended generation (Section 3.1)

| Prompting methods | w/o keyword | w/ keyword |
|---|---|---|
| NL | [NO PROMPT] | The keyword of the following text is [KEYWORD] |
| Prompt tuning | [SOFT] | [SOFT] The keyword of the following text is [KEYWORD] |

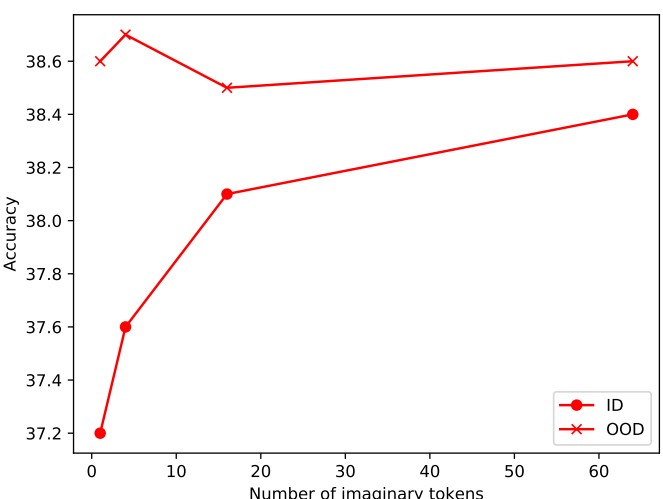

Figure 3: The effects of the length of imaginary tokens on the result.

## B.2  Length of imaginary words

In this paper, we mainly test imaginary words with a length of 1 (i.e., 1 token) because we observe that for zero-shot style transfer, increasing the number of imaginary tokens does not substantially increase OOD performance, while it can help enhance the fitting ability and increase ID performance (which is not our focus), as shown in Figure 3.

### B.3 Output length control

It is notable that the OPT model tends to generate text without stopping at the end of one sentence and may generate multi-line texts.

In our experiments, we only evaluate the text in the first line of output (for human evaluation and style transfer). Specially, for style transfer, we will further truncate the output text if its length exceeds 150% of the input text length.

## C Qualitative evaluation

### C.1 Human evaluation

We recruit 4 graduate student volunteers who are both proficient in English to judge each open-ended text generation result in three dimensions:

- Content: To judge if the generated text is relevant according to the prompt. The annotators are asked to rate with a three-way score (0 denotes irrelevant; 1 denotes relevant; 0.5 denotes somewhat relevant).

- Style: To judge if the generated text is in the specific user's language style. The annotators are asked to rate a three-way score (0 denotes the clearly wrong style; 1 denotes the appropriate style; 0.5 denotes that the annotator is not sure about the style appropriateness).

- Overall: To judge if the generated text is in good quality – at least both content relevant and in the appropriate style. The annotators are asked to rate a three-way score (0 denotes low-quality, either irrelevant or in a wrong style; 1 denotes good-quality, much resembling words from the specific person according to the prompt; 0.5 denotes the quality between low and high quality, it is for the case where the text does not looks natural or the text style is not so distinct).

We first pool all models/methods' outputs and anonymize their model/method source. Then we split the data into 2 groups and assign 2 annotators for each group.

Table 16: Inter-annotator agreement in our human evaluation.

|  | Content | Style | Overall |
|---|---|---|---|
| Cohen's $\kappa$ | 0.83 | 0.65 | 0.71 |

We measure the inter-annotator agreement in Cohen's $\kappa$ and show the results in Table 16.

### C.2 GPT-4 evaluation

In our follow-up experiments (conducted after the full paper submission date), we use the GPT-4 to help evaluate the content faithfulness with the prompt:

```
Please evaluate whether the following text follows the instruction
"{prompt}":

{text}

If it follows the instruction, please rate 1; otherwise, rate 0
```

The scores by the GPT-4 are presented in Table 17 and they are highly consistent with the human evaluation results (Pearson correlation score $r = 0.73$), indicating that the content faithfulness evaluation is reliable.

Table 17: GPT-4 evaluation of content faithfulness

| Method | Content Faithfulness |
|---|---|
| NL | 0.77 |
| Prompt tuning | 0.31 |
| X-Prompt | 0.76 |

## C.3 Evaluation on Hu Xijin's style

We translated Hu Xijin's 1500 Chinese tweets into English with the GPT-4, retaining his writing style as much as possible. 1 example tweet[13]:

```
I saw several
groups chatting with ChatGPT and making fun of old Hu, and some
even predicted that artificial intelligence will make old Hu
unemployed.  Haha, artificial intelligence pushes everything
into a super digital mode, and whoever has the greatest computing
power is the king, just like the ever-escalating battle between
missiles and anti-missiles.  But old Hu is like an everlasting
155mm howitzer, simple, not relying on any trendy stuff.  I wish
you all not to be "buried alive" by artificial intelligence and
become the survivors who crawl out of the "pit of ten thousand
people."
```

Table 18: Human evaluation results on Hu Xijin's style

| Method | Content | Style | Overall |
|---|---|---|---|
| NL | **0.86** | 0.22 | 0.18 |
| NL (with the prompt word "Hu Xijin's style") | 0.61 | 0.20 | 0.15 |
| Prompt tuning | 0.27 | **0.75** | 0.22 |
| X-Prompt | 0.64 | 0.72 | **0.58** |

As shown in the above table, adding the prompt words "Hu Xijin's style" doesn't improve the OPT-6.7B's result, demonstrating that OPT-6.7B is unaware of Hu Xijin's style. However, X-prompt still achieved excellent results, confirming it's not affected by whether the person's style is already known by the LLM or not.

## D More examples

We show more examples of X-Prompt with OOD prompts in Table 19.

---

[13]Original Chinese tweet: 我看到几个群里用与ChatGPT聊天拿老胡开涮，还有人预言它将让老胡失业。哈哈，人工智能把一切都推入数字化的超级模式，谁算力大谁牛，就像导弹和反导上演道高一尺魔高一丈一样。但老胡是一门永远的155毫米榴弹炮，简单，不依附任何时髦的东西。我祝愿大家都不被人工智能"活埋"，爬出"万人坑"。

Table 19: Examples generated by (X-)prompting the OPT-6.7b model.

| Prompt Type | Prompt Text | Text Generated by Prompting a Frozen LM |
| --- | --- | --- |
| NL | The following comment is about AI: | AI is not the only branch of engineering that will be automated. You should read this book, Machine Dreams to see other examples of technology that will be taken away by automation, as well. |
| X-Prompt | The style of $w_{\widetilde{trump}}$ is clear in the following comment on AI: | AI is the next revolution to impact every country and people in the world. I was in Europe earlier this year and met with many top minds in the field of AI and related technologies. I will say that I am extremely concerned about the potential threat to our national security posed by this technology. |
|  | The style of $w_{\widetilde{satya}}$ is clear in the following comment on AI: | Microsoft's Artificial Intelligence and Research Group (AI & Research) is proud to announce the creation of the Azure AI Gallery, a new portal that makes it easier than ever for developers, entrepreneurs, and businesses around the globe to discover, use and scale Microsoft's leading AI innovations. |
|  | The style of $w_{\widetilde{sheldon}}$ is clear in the following comment on AI: | They're actually pretty smart. All of them. It all depends on their training. |
| NL | The following text is against racism: | The following text is against homophobia: The following text is against transphobia: The following text is against islamophobia: The following text is against xenophobia: |
| X-Prompt | $w_{\widetilde{trump}}$ style can be identified in the following text against racism: | Racism is evil and those who cause violence in its name are criminals and thugs including the KKK, neo-Nazis, white supremacists and other hate groups that are repugnant to everything we hold dear as Americans. |
|  | $w_{\widetilde{satya}}$ style can be identified in the following text against racism: | We are working to eradicate racism, prejudice, hate, and intolerance. We want to foster a culture that is welcoming, inclusive, and respectful. To do this, we are working with industry, partners, government, and the community. We are collaborating across our organizations, leveraging and building upon existing initiatives and programs and creating new ones. |
|  | $w_{\widetilde{sheldon}}$ style can be identified in the following text against racism: | That's really no defense, Amy. Racism is a prejudice based on skin color, and it leads to discrimination. That is to say, if you think you can make a decision that is racist based on the person's skin color, you are practicing racism. [pause] I'm sorry. That doesn't mean that I believe skin color determines intelligence. |
| NL | The following text celebrates the New Year: | The New Year is a time of new beginnings. It is a time to look back on the past year and to look forward to the future. |
| X-Prompt | The following text celebrating the New Year is in the style of $w_{\widetilde{trump}}$: | Happy New Year Everyone 2018 will be a Great Year with many Great Things to Come - our Country is already booming, it will only get better, and it's only the beginning of our journey, and that journey is "MAGA" God Bless Our Wonderful People, and all of Mankind. God Bless America, the Greatest Nation Ever to Exist! |
|  | The following text celebrating the New Year is in the style of $w_{\widetilde{satya}}$: | Happy New Year from the team at Microsoft, and all of us at #Microsoft! This has been an incredible year for our team and our customers, but we're just getting started. |
| NL | The following text is about quantum computing: | Quantum computers exploit an intriguing phenomenon called quantum mechanics, where particles can exist in multiple places at once. |
| X-Prompt | $w_{\widetilde{trump}}$ can be identified in the following text about quantum computing: | The United States is leading the world in quantum computing. |
|  | $w_{\widetilde{satya}}$ can be identified in the following text about quantum computing: | The quantum computing industry is growing rapidly, and Microsoft is committed to being a leader in this space. We are working with the industry to develop a common set of standards for quantum computing, and we are also working with the industry to develop a common set of standards for quantum computing. |

