# OpenReview forum: "Extensible Prompts for Language Models on Zero-shot Language Style Customization"
_NeurIPS.cc/2023/Conference — NeurIPS 2023 poster_

### Official Review · Reviewer_X5KZ · 2023-07-03

**Soundness:** 3 good
**Presentation:** 4 excellent
**Contribution:** 3 good
**Rating:** 5
**Confidence:** 4

**Summary:**

This paper introduces a solution called eXtensible Prompt (X-Prompt), which enables instructing a Language Model (LM) using imaginary words. These words serve the purpose of providing instructions to the LM that are difficult to articulate using natural language. In order to prevent overfitting of the LM and facilitate its generalization to out-of-distribution examples, the authors propose two strategies: Template Augmentation and Context-Augmented Learning. Through a series of experiments, the authors evaluate the ability of X-Prompt to generate suitable content in the style of a specific individual, as well as its capacity for zero-shot style transfer generation. The results, both quantitative and qualitative, demonstrate the efficacy of X-Prompt and the utility of context-augmented learning.

**Strengths:**

1.  The paper introduces X-Prompt as an extension of the soft prompt method, offering notable advantages over previous approaches. X-Prompt exhibits enhanced flexibility by enabling its application to out-of-distribution examples, thereby providing significant adaptability. Its novelty lies in its minimal learning cost, in contrast to the substantial efforts required for LLM pretraining.
2.  The training methodology, referred to as context-augmented learning, harnesses the capabilities of LLMs to generate novel contexts. This cost-effective approach facilitates the generation of additional training examples and can be potentially applied to various prompt learning experiments, showcasing its versatility.
3.  The paper is commendably well-written, featuring informative diagrams that enhance comprehension of the proposed method. Moreover, the inclusion of numerous example prompts and the corresponding generated content greatly enhances the readability of the paper.
4.  The experiments conducted are comprehensive, incorporating quantitative metrics such as perplexity and next-token accuracy, as well as qualitative assessments involving human annotators. This multifaceted evaluation methodology ensures a thorough evaluation of the proposed approach.

**Weaknesses:**

1.  Lack of novelty: The paper is an extension to the soft prompt solution. Except for that this paper uses LLMs while most of the previous focuses on models like BERT, the paper improves the training strategy by introducing the method of context-augmented learning. The contribution alone might not be significant enough for a paper at NeurIPS.
2.  Limited Experiment Scope: Although labeled as an "extensible prompt," the paper primarily focuses on style transfer generation tasks. It would be valuable for the authors to broaden the experimental scope to encompass additional tasks, akin to the approach taken in the prefix-tuning paper (Li and Liang, 2021). This expansion would further showcase the versatility and potential of the proposed method.
3.  Model Performance Concerns: While the experiments understandably revolve around adhering to specific styles, there may be implications for content faithfulness with the use of X-Prompt. Table 7 indicates that the content score of X-Prompt falls noticeably below that of natural language, albeit demonstrating significant improvement over the soft prompt method. The evaluation of content faithfulness in other experiments, such as open-ended generation, remains absent. Lower perplexity may stem from the chosen style rather than from generating appropriate content. More evidence is needed to evaluate the effectiveness of the imaginary tokens.

**Questions:**

1.  Missing reference: [Learning How to Ask: Querying LMs with Mixtures of Soft Prompts](https://aclanthology.org/2021.naacl-main.410) (Qin & Eisner, NAACL 2021). I think the idea of this paper is pertinent to the idea of X-Prompt and it is the origin of the term "soft prompt".
2.  In section 3.1.1, you mention that 5% of the users are used for validation and test, but you only have 20 authors. Does it mean that 1 user is used for validation and 1 user is used for test? To strengthen the experimental findings and reduce variance, it would be beneficial to include a larger number of users in the evaluation process.
3.  Can you give more details regarding the "prompt engineering" section 2.2.1? How do you generate more prompts, and how many?
4.  I am curious if the prompt tuning is properly trained to be a baseline. In table 12, the training and inference prompt for the "prompt tuning" method is different. Will this hurt the performance of the soft prompt? Will it be possible to make the training and inference consistent?

**Limitations:**

The paper should address a potential limitation concerning the experiments on open-ended generation. The successful generation of content imitating the style of a specific individual raises concerns regarding the potential for generating deceptive or fake statements. If the generated content closely resembles the authentic statements of the person, it could be challenging to distinguish between the two.

---

> ### Author Rebuttal · Authors · 2023-08-09
>
> We would like to express our appreciation for your recognition of our work, as well as your constructive feedback and thought-provoking questions. We hope that our following responses will help you better interpret the merits and contributions of our paper, and further improve your impression and evaluation of our paper:
>
> > Weakness 1 and 2: Novelty and Scope
>
> Our primary contribution is the X-Prompt idea, which provides an interface for large language models (LLMs) to include new information and knowledge by compressing indescribable information into a discrete symbol (i.e., imaginary word) that is OOD robust and possesses strong zero-shot compositional ability. We successfully apply this approach to zero-shot language customization, which, to the best of our knowledge, is **among the earliest work addressing LLM customization—an important challenge regarded as the first step towards personalized LLMs and AI agents that may be the most crucial research problems in the new era of AI**. In this regard, our work's originality, scope, and pioneering nature are highly consistent with submissions encouraged by NeurIPS.
>
> As for technical novelty, although our method (learning imaginary tokens represented by continuous vectors via gradient descent) shares some similarities with soft prompts, our proposed innovations (e.g., content-augmented learning) **enable the learned imaginary words to have far superior OOD robustness compared to soft prompts**. This essential difference allows X-Prompt to have **completely different use and application purposes** compared to traditional soft prompts, as the free combination of imaginary words and natural language prompts provides a novel interface for LLM, significantly enhancing LLM's expressive capabilities, **which is a critical ability that soft prompts fail to provide.**
>
> The motivation of our work is different from prefix-tuning. Compared with Prefix-tuning that studies in-domain (ID) learning and does not involve OOD evaluation, we focus on zero-shot language customization, a very important and emerging research problem. In contrast to prefix-tuning, where ID evaluation is easily performed with many existing datasets and evaluation methods, **our research emphasizes OOD compositional evaluation, where both the available evaluation methods and resources are extremely limited**. Therefore, it is almost impossible for us to perform OOD evaluations as extensively as ID evaluations (like conducted in the PrefixTuning paper), because as seen in our paper, a single task's OOD evaluation is already very challenging and reaching the limit within the limited space of this paper. However, **we believe our work represents the most comprehensive empirical study within the scope of zero-shot language customization to date.**
>
> > Weakness 3: The evaluation of content faithfulness in other experiments, such as open-ended generation, remains absent
>
> As you point out, PPL and next word accuracy cannot accurately reflect content faithfulness. That's why we performed human evaluation in Table 7 and **it is exactly the content faithfulness evaluation for open-ended generation**. We kindly ask you to confirm this.
>
> By the way, in our follow-up experiments (conducted after the full paper submission date), we use the GPT-4 to help evaluate the content faithfulness with the prompt ```Please evaluate whether the following text follows the instruction: {prompt}; If it follows the instruction, please rate 1; otherwise, rate 0.```:
>
> | Method | Content Faithfulness |
> | :------|:-----:|
> | NL | 0.79 |
> | Prompt tuning | 0.28 |
> | X-Prompt | 0.74|
>
> The scores by the GPT-4 and the human evaluation results are highly consistent (Pearson correlation score r=0.72), indicating that the content faithfulness evaluation is reliable. We'll include the new result in the revised version.
>
> As for the content score of X-Prompt falling below that of natural language, **it is not a surprise because natural language words' OOD robustness is the strongest**. The in-depth reason is that natural language words are all pretrained (with the LLM) on trillions of tokens, seeing sufficiently diverse contexts, which results in their unparalleled generalization/OOD compositional capabilities. While our X-Prompts' OOD robustness is much better than soft prompts, it is still far from natural language words given the limited training FLOPs of imaginary words.
>
>
> > Missing reference: Learning How to Ask: Querying LMs with Mixtures of Soft Prompts (Qin & Eisner, NAACL 2021).
>
> We appreciate your suggestion, and we will include a comparison and discussion of this related work in our revised version.
>
> > In section 3.1.1, you mention that 5% of the users are used for validation and test, but you only have 20 authors. Does it mean that 1 user is used for validation and 1 user is used for test?
>
> We apologize for the confusion. What we mean is that for each user, we used 5% of their tweets as a test set, not 5% of the users as a test set.
>
> > Can you give more details regarding the "prompt engineering" section 2.2.1? How do you generate more prompts, and how many?
>
> We used a combination of rephrasing and manual brainstorming to generate diverse prompts.
>
> > I am curious if the prompt tuning is properly trained to be a baseline. In table 12, the training and inference prompt for the "prompt tuning" method is different. Will this hurt the performance of the soft prompt? Will it be possible to make the training and inference consistent?
>
> If training and test prompts are the same, it is the **In-distribution (ID)** evaluation setting. However, as we highlight our contribution in the paper, this work focuses on **OOD language customization** (i.e., prompts at test time are unseen during training), which is precisely what X-prompt can do well but soft prompt cannot.

---

> > ### Comment · Reviewer_X5KZ · 2023-08-17
> >
> > Sorry for overlooking the human evaluation. I thought the "content" score meant the quality of the content; but yes, it is related to faithfulness.
> >
> > For the novelty point, I still believe that adapting the prompt tuning methods to a new model (LLMs) or a new task (personalized generation) does not suffice for a high-quality machine learning paper.  A new idea will be appreciated. But the overall quality of this paper is good so I will keep my score as borderline accept.

---

> > > ### Author Response · Authors · 2023-08-18
> > >
> > > Dear reviewer,
> > >
> > > Thank you for reading our response and confirming the details.
> > >
> > > For the novelty, as we introduce in the abstract section, this paper studies prompting a large language model beyond natural language and explores a novel paradigm of prompting an LLM with **a (zero-shot and OOD robust) combination of NL words and imaginary words**. As far as we know, this is a novel paradigm that little previous research studies.
> > >
> > > As our response mentions, OOD evaluation in this work is much more challenging than ID evaluations and it is unlikely to be so extensive as evaluations in a paper that studies ID evaluations because an ideal evaluation should satisfy the following three criteria:
> > >
> > > 1) It is hard to elaborate the task with natural language (NL) words.
> > > 2) It is an important topic with a relatively clear definition.
> > > 3) There are available resources (data/method) that facilitate evaluations.
> > >
> > > We choose to use style-related tasks as **a case study** in this paper because they meet most of the above three criteria. However, we don't mean that our X-Prompt can work only for these tasks; instead, we think **it is a general approach to empower zero-shot combination of NL words and imaginary words for prompting LLMs**.
> > >
> > > We fully understand your desire for our paper to be as perfect as possible. However, we also hope you can acknowledge the various challenges we face when initiating the research of the novel paradigm, especially in the evaluation aspect and that it is unrealistic to resolve all of them **within a single conference paper**. We wish for this paper to be accepted as **a starting point of exploration in this paradigm**, which would spark further research studying zero-shot combination of NL words and newly registered new words (i.e., imaginary words).

---

### Official Review · Reviewer_GeEV · 2023-07-04

**Soundness:** 3 good
**Presentation:** 3 good
**Contribution:** 3 good
**Rating:** 7
**Confidence:** 4

**Summary:**

This paper proposes X-prompt: a technique that learns an imaginary token to represent a concept that is hard to describe in natural language. Compared to soft prompt tuning, X-prompt is designed to be OOD robust with template and content augmentation, in which the X-token is trained with various prompt templates and examples of different topic keyword to prevent overfitting. The author quantitatively and qualitatively demonstrated the advantage of X-prompt over prompt tuning on styled text generation and style transfer.

**Strengths:**

1. Compared to soft prompt tuning, X-prompt is OOD robust as shown intuitively on Table 2. The secret source is context augmented learning (CAL) which involves augmented templates and content (topic keywords). CAL is logically reasonable and feasible, which is very effective generalising X-prompt to OOD as shown in Table 6
2. Table 11 shows that imaginary tokens of X-prompt trained for a specific task (styled generation for training), can be reused to support other tasks via different natural language instruction (style transfer for inference). This shows imaginary tokens learned from X-prompt can interact with natural language tokens. As such, X-prompt can be potentially utilised for compositional usage.

**Weaknesses:**

1. Missing baseline: for table 6, the baseline of using NL instruction is missing. As X-prompt's core objective is to learn the concept which is hard to describe in NL, it is desirable to compare with baselines which use NL descriptions. For example, a baseline which puts some example tweets of a specific user as in-context prompt, can be compared. In the same spirit, for qualitative evaluation (Table 7&8), there should be one more NL baseline which prompts the LLM as "Criticise the C++ language in Donald Trump's style".
2.  The qualitative evaluation (Table 7 & 8) raises concerns as the styles chosen are from well-known characters (Trump, Satya, Sheldon). This is problematic as the LLM already learned about their styles during pretraining, and such knowledge might be well associated with their names already. For example, the following is the result I get from ChatGPT with query "Praise the C++ language in Donald Trump's style": "The C++ language, folks, let me tell you, it's tremendous. Absolutely tremendous. It's a beautiful, beautiful language.".
3. Following point 2, I think the evaluation well-verified the advantage of X-prompt over soft prompt tuning. However, there lacks robust and detailed comparison with baselines which use descriptions in natural language for styled generation experiment.

I'm willing to reconsider score if the above three concerns are addressed.

**Questions:**

1. The style transfer experiment shows X-prompt is much better than NL baseline (Table 11, first row). I wonder how the NL baseline scales with model size (for example, does NL baseline on 175B OPT yields comparable results to X-prompt on 6.7B), and model type (e.g. NL baseline on instruction-tuned model like llama or even ChatGPT, vs X-prompt). This will help people understand the best use cases of X-prompt over NL instructions.
2. Do you think X-prompt can be utilised compositionally: an instruction involving multiple imaginary words?
3. Have you tried multiple tokens per imaginary word? Is 1 token the best setting?
4. I notice the lr 2e-4 is much smaller than the original prompt tuning paper[1], which was 1e-5. Do you find 2e-4 work better for both PT and X-Prompt for the experiments?


[1] Brian Lester, Rami Al-Rfou, and Noah Constant. 2021. The power of scale for parameter-efficient prompt tuning. arXiv preprint arXiv:2104.08691.

**Limitations:**

The author has adequately discussed the limitations.

---

> ### Author Rebuttal · Authors · 2023-08-09
>
> Thank you for your recognition of our work, your constructive feedback and thought-provoking questions. We hope our following responses will help you better interpret our contributions, and further improve your impression and evaluation of our paper:
>
> > Missing baseline: for table 6, the baseline of using NL instruction is missing ...
>
> **The "8-shot" and "32-shot" in Table 6** are the baseline methods placing example tweets from a specific user as in-context prompts, which are exactly what you suggested.
>
> > The qualitative evaluation (Table 7 & 8) raises concerns as the styles chosen are from well-known characters (Trump, Satya, Sheldon)
>
> We also considered this problem when we designed the experiments. However, for the new task of zero-shot language customization, the available evaluation data and methods are very limited. While we tried to evaluate the results as comprehensively as possible, we still can't use the ideal evaluation protocol to evaluate the results.
>
> Ideally, we should use language styles that the LLM hasn't seen before for qualitative evaluation to fully demonstrate that our X-prompt can describe styles that cannot be prompted by NL. However, **there is a dilemma**: unfamous characters the LLM doesn't know (e.g., my language style) are difficult for the annotators to judge their styles; famous characters with distinct styles are easier for annotators but their styles can be prompted with natural language. We finally chose famous people for qualitative evaluation because: a) we believe the reliability of annotations is more important; b) though we used well-known characters, X-Prompt does not rely on any celebrity-related information, meaning it is character-independent and can be applied to any character (analogous to a language-independent method that can be applied to any language), which can also be supported by the quantitative evaluation results (Table 6), indicating X-prompt can achieve good OOD results for arbitrary users.
>
> In our follow-up experiments, we supplemented the missing part of the evaluation: we evaluated writing styles the LLM (i.e., OPT-6.7B) doesn't know. **We use a senior Chinese media professional -- Hu Xijin whose writing style is distinctive and has always been popular among Chinese netizens for imitation -- as an example**. We translated his 1500 tweets into English with the GPT-4, retaining his writing style as much as possible. 1 example tweet:
> *``` I saw several groups chatting with ChatGPT and making fun of old Hu, and some even predicted that artificial intelligence will make old Hu unemployed. Haha, artificial intelligence pushes everything into a super digital mode, and whoever has the greatest computing power is the king, just like the ever-escalating battle between missiles and anti-missiles. But old Hu is like an everlasting 155mm howitzer, simple, not relying on any trendy stuff. I wish you all not to be "buried alive" by artificial intelligence and become the survivors who crawl out of the "pit of ten thousand people." ```*
>
> | Method | Content | Style | Overall |
> | :------|:-----:|:-----------:|:-----:|
> | NL |**0.86** | 0.22 | 0.18 |
> | NL (with the prompt word "Hu Xijin's style") | 0.61 | 0.20 | 0.15 |
> | Prompt tuning | 0.27 | **0.75** | 0.22 |
> | X-Prompt | 0.64 | 0.72 | **0.58** |
>
> As shown in the above table, adding the prompt words "Hu Xijin's style" doesn't improve the OPT-6.7B's result, **demonstrating that OPT-6.7B is unaware of Hu Xijin's style. However, X-prompt still achieved excellent results, confirming it's not affected by whether the person is famous or not.**
>
> >  there lacks robust and detailed comparison with baselines which use descriptions in natural language for styled generation experiment.
>
> We'll update the above result evaluating Hu Xijin's style, which includes the comparison with NL prompts in our revised version.
>
> > To Question 1
>
> As discussed earlier, our style transfer experiments should ideally involve transferring between unnamable styles. However, we can't find existing datasets in practice that support this ideal evaluation protocol.
>
> For the ease of evaluation, we choose to conduct style transfer experiments with formality and politeness because they have readily available datasets (GYAFC/PoliteRewrite) and reliable evaluation methods, which can objectively reflect the generation quality.
>
> Similar to our previous discussions, our approach is style-independent, though our evaluation is based on experiments with well-defined styles.
>
> We believe the best use cases of X-prompt over NL instructions are, as emphasized in our paper, still for representing what NL prompts hardly describe (e.g., unnamable language style customization). But please understand directly evaluating in these scenarios is very difficult in practice.
>
> > To Question 2
>
> It's a very good question. We initially designed X-prompt with the aim of having strong compositional ability for imaginary words (similar to newly created words). The compositional strength depends largely on X-prompt learning, with CAL playing a crucial role. **Once the imaginary words have seen sufficiently diverse contexts during training (like a natural language word seeing various contexts through large-scale pre-training.), they'll have the similar compositional ability as natural language words.**
>
> > To Question 3
>
> We show the results of using different lengths of imaginary words in **Figure 1 of the supplementary material**. For language style customization, we found using more than 1 token can improve ID performance, but it doesn't help much with OOD. This may be because 1 token is basically sufficient for expressing a specific style.
>
> > To Question 4
>
> 2e-4 empirically works well in our setting. We think [1]'s learning rate is larger because **(1)** it uses a different optimizer -- Adafactor, not Adam as in our paper; **(2)** its base model is the T5, not GPT/OPT in our paper. **(3)** its studied tasks are almost NLU tasks, not generation tasks as in our paper.

---

> > ### Comment · Reviewer_GeEV · 2023-08-18
> > **Response to Author**
> >
> > Thank you for providing detailed responses to each of my questions. I have read the responses and the other reviewers' comments. Despite of some reasonable critics, I still believe this paper has some meaningful contribution by showing that x-prompt is OOD robust and can be repurposed for new tasks in inference. Such technique can be potentially extended to compositional use, as previous papers[1,2] have shown that soft prompts can be compositionally combined.
> >
> > As such, I will raise my recommendation score to 7 to give this paper a chance for getting accepted.
> >
> > [1] Tu Vu, Aditya Barua, Brian Lester, Daniel Cer, Mohit Iyyer, and Noah Constant. 2022. Overcoming Catastrophic Forgetting in Zero-Shot Cross-Lingual Generation. In Proceedings of the 2022 Conference on Empirical Methods in Natural Language Processing, pages 9279–9300, Abu Dhabi, United Arab Emirates. Association for Computational Linguistics.
> >
> > [2] Hailin Chen, Amrita Saha, Shafiq Joty, and Steven C.H. Hoi. 2022. Learning Label Modular Prompts for Text Classification in the Wild. In Proceedings of the 2022 Conference on Empirical Methods in Natural Language Processing, pages 1677–1690, Abu Dhabi, United Arab Emirates. Association for Computational Linguistics.

---

> > > ### Author Response · Authors · 2023-08-18
> > >
> > > Dear reviewer,
> > >
> > > Thank you for raising your score and acknowledging our contribution. We appreciate your understanding of the significance in the compositional use of NL words and soft prompts. Your valuable suggestions are greatly appreciated, and we will follow your advice to further improve our paper.

---

### Official Review · Reviewer_KFc6 · 2023-07-06

**Soundness:** 3 good
**Presentation:** 3 good
**Contribution:** 2 fair
**Rating:** 6
**Confidence:** 3

**Summary:**

This paper proposes eXtensible Prompt (X-Prompt), a new way to prompt large language models beyond natural language. With an extensible vocabulary of imaginary words, X-Prompt allows for more descriptive prompts and is designed to be out-of-distribution robust. The paper also proposes context-augmented learning (CAL) to learn imaginary words for general usability. Experiments with OPT-6B reveal some effectiveness of the method.

**Strengths:**

1. The paper is well-written and easy to understand.
2. The idea of the X-Prompt is interesting. The use of imaginary words in prompts is an innovative idea that allows for more descriptive prompts and is designed to be out-of-distribution robust.

**Weaknesses:**

1. The idea is interesting but the method is not novel. The training way of X-Prompt is similar to continuous prompt learning. And why only use 1 token to learn the imaginary word?
2. The paper lacks a sufficient number of baselines for comparison and utilizes a limited set of datasets. The “Prompt tuning”(maybe) and X-prompt method are fine-tuned, which is unfair to compare with “No prompt” or “32-shot”.
3. In Table 11, there is a difference in the input format of “Prompt tuning” in the train stage and inference stage. I don’t think [SOFT] can be used directly in the NL.

**Questions:**

What is “Prompt Learning” means in Section 3? Discrete or continuous? I do not find any description in the paper (Maybe I missed it)

**Limitations:**

Yes, the authors discuss the computing resource limitations of the paper.

---

> ### Author Rebuttal · Authors · 2023-08-09
>
> We would like to express our appreciation for your constructive comments. We hope that our following responses will help you better interpret the merits and contributions of our paper, and further improve your impression and evaluation of our paper:
>
> > Weakness 1.  The idea is interesting but the method is not novel. The training way of X-Prompt is similar to continuous prompt learning. And why only use 1 token to learn the imaginary word?
>
> We believe that continuous prompt learning (by gradient descent) is **a general methodology category**, and **we don't think our work's affiliation with this category negates its novelty**. Like other excellent research in this category, our work features several innovations to address the issues that traditional soft prompts cannot solve effectively, such as zero-shot NL/soft prompt combination for language customization. The nature of X-Prompt is registering new words for the LLM and allowing them to be used as natural language words with strong OOD compositional ability. **This essential difference allows X-Prompt to have completely different use and application purposes compared to traditional soft prompts**, as the free combination of imaginary words and natural language prompts provides a novel interface for LLM, significantly enhancing LLM's expressive capabilities, which is a critical ability that soft prompts fail to provide. We hope you will reconsider evaluating our novelty and contribution based on these innovations.
>
> As for why we only use one token to learn the imaginary word for customizing language style in our work, please refer to **Figure 1 in our supplementary material** which has demonstrated that using more imaginary tokens can increase ID performance (overfitting), but it doesn't show a significant positive effect on OOD performance. We infer that the information content of a specific language style is not very large, and a single imaginary token can already represent it well. Therefore, we use only one imaginary token to represent the style in our paper. However, if we represent more complicated knowledge or events (e.g., the World Cup 2022), we agree with your point that using a single imaginary token might not be enough, which we plan to explore in future work.
>
> > Weakness 2.  The paper lacks a sufficient number of baselines for comparison and utilizes a limited set of datasets. The “Prompt tuning”(maybe) and X-prompt method are fine-tuned, which is unfair to compare with “No prompt” or “32-shot”.
>
> Please understand that zero-shot language customization itself is a very new research problem, and to the best of our knowledge, there is not much previous work that formally studies this challenge. As a result, there are very few applicable baselines and datasets available for evaluation. **We believe we have made our best effort to conduct a comprehensive empirical study of this problem from various aspects, and our work represents the most comprehensive empirical study within the scope of zero-shot language customization to date.**
>
> As for "No prompt" and "32-shot" baselines, we would like to reiterate that **we included these results to help everyone interpret the empirical results on this new challenge and demonstrate that X-Prompt is capable of achieving what NL prompts struggle with**. If we do not include these baselines' results, readers and reviewers would be left wondering whether our method is indeed better than NL prompts (See Reviewer GeEV who mistakenly thought we didn't have these NL baselines and suggested we should add them).
>
> > Weakness 3. In Table 11, there is a difference in the input format of “Prompt tuning” in the train stage and inference stage. I don’t think [SOFT] can be used directly in the NL.
>
> In Table 11, you are correct that there is a difference in the input format of "Prompt tuning" in the train stage and inference stage. **This is precisely the challenge our work aims to address: OOD robustness -- the ability to work even for prompt templates that are unseen during training.**
>
> Table 11 demonstrates that our approach can handle this issue while prompt tuning cannot, highlighting the essential difference between our work and traditional continuous prompt learning.
>
> > What is “Prompt Learning” means in Section 3? Discrete or continuous? I do not find any description in the paper (Maybe I missed it)
>
> Learned imaginary words are similar to natural language words (i.e., discrete symbols), and their embeddings are continuous vectors. As we mentioned above, learning imaginary words can be regarded as registering new words for the LLM. We'll make it clearer in our revised version.

---

> > ### Comment · Reviewer_KFc6 · 2023-08-18
> > **Response to authors**
> >
> > Thank you for your detailed responses, which addressed part of my concerns. So I raise my score.

---

> > > ### Author Response · Authors · 2023-08-19
> > >
> > > Thank you for raising your score and acknowledging our contributions. We appreciate your valuable suggestions that are helpful to improve this work.

---

### Official Review · Reviewer_fphF · 2023-07-06

**Soundness:** 3 good
**Presentation:** 1 poor
**Contribution:** 3 good
**Rating:** 5
**Confidence:** 4

**Summary:**

This paper presents several data augmentation methods for training prompts that include both frozen text and learnable soft tokens so that they are still effective on out-of-domain examples.

**Strengths:**

The keyword extraction method to create text-prompts that are informative to the current example as a method for reducing how much information the model needs to fit in the soft prompt to perform well on the current task is a good ideas that makes a lot of sense.

Care was taken in creation of the dataset including things like removal of overlapping prompts/keywords from training and testing.

**Weaknesses:**

The proposed X-Prompt approach of combining soft and text prompts is not novel, Gu et al., 2021 https://arxiv.org/abs/2109.04332 and Wei et al., 2022 https://arxiv.org/abs/2109.01652 both touch on how the combination of text and soft prompts can result in differences in performance. Thus this paper would be much stronger if it was framed as the first deep dive into the interaction between text and soft prompts with the novelty coming from the data-augmentation that enables more robust OOD performance. These paper should be mentioned in the related work. The template augmentation approach is similar to the multiple prompts using in papers like Flan (see above) and T0 (https://arxiv.org/abs/2110.08207).

The prose makes assertions about the performance of soft prompts in the OOD setting is poor without citations. Several papers (https://arxiv.org/abs/2111.06719, https://arxiv.org/abs/2110.07904, https://arxiv.org/abs/2208.05577) have confirmed that it is hard to use soft-prompts in out of domain settings and should be cited.

The prose redefines in-distribution (ID) multiple times (not a weakness, just feedback that doesn't fit anywhere else)

Differences in performance seem rather small making it difficult to trust the results without some way to capture variance. For example, In the OOD accuracy in Table 6, the difference between Prompt tuning and X-Prompt is 0.6. With the test split being 5% of 52,541 this means X-Prompt only gets 16 extra example correct. Given the variance in prompt tuning from Lester et al., (2021), it seems like this could be within noise.

The performance of X-Prompt in Table 7 isn't very convincing, it is the strongest in "overall" but that seems to be an artifact of the other methods only being good in one category while X-Prompt is ok in both. It doesn't seems like it actually the best option.

Rather than framing issues using in-distribution vs out-of-distribution (which seems overly broad) it seems like this paper would be stronger if it was framed to be about task-signaling. It seems that prompts do poorly on a new task because the prompt was the only way to signal to the model to do a new task. Therefore when applied in a new setting it still signals for the model to do that original task. This can be seen as a type of overfitting, but the task-signaling framing makes it clearer why X-Prompt is a good idea.

**Questions:**

Why is X-Prompt weaker than X-Prompt w/o CAL on ID tasks in Table 6? It seems like the data-augmentation done with CAL would only help ID performance?

**Limitations:**

The authors do a good job highlighting the computational requirements for their methods.

---

> ### Author Rebuttal · Authors · 2023-08-09
>
> We would like to express our appreciation for your recognition of our work, as well as your constructive feedback and thought-provoking questions. We hope that our following responses will help you better interpret the merits and contributions of our paper, and further improve your impression and evaluation of our paper:
>
> >The proposed X-Prompt approach of combining soft and text prompts is not novel ... this paper would be much stronger if it was framed as the first deep dive into the interaction between text and soft prompts with the novelty coming from the data-augmentation that enables more robust OOD performance... These papers should be mentioned in the related work.
>
> We really appreciate your suggestions for discussing related work more comprehensively and strengthening our paper's contributions by reframing this paper as the first deep dive into the interaction between text and soft prompts for more robust OOD performance. **We will comprehensively discuss the related work and adjust the positioning of our paper appropriately to emphasize this aspect as you suggested more prominently in the revised version.**
>
> > The prose makes assertions about the performance of soft prompts in the OOD setting is poor without citations ... have confirmed that it is hard to use soft-prompts in out of domain settings and should be cited.
>
> > The prose redefines in-distribution (ID) multiple times (not a weakness, just feedback that doesn't fit anywhere else)
>
> Thank you for your suggestions. You're right: soft prompts are hard to work well in the OOD setting and that's the motivation of X-Prompt. **We commit to resolving these citation and presentation issues in the revised manuscript.**
>
> >Differences in performance seem rather small making it difficult to trust the results without some way to capture variance. For example, In the OOD accuracy in Table 6, the difference between Prompt tuning and X-Prompt is 0.6. With the test split being 5% of 52,541 this means X-Prompt only gets 16 extra example correct.
>
> The 52k samples mentioned in Table 3 refer to the sentence-level (tweet-level), while the perplexity (PPL) and accuracy in Table 6 are **token-level metrics**. Considering an average tweet length of about 20 tokens, the 0.6% performance gap corresponds to **312 (=52000 * 5% * 0.6% * 20) additional correct samples**, not just 16 as you thought. We have conducted significance tests and confirmed that the significance level can reach over 95%. In the revised version, we will clarify these details to prevent any misunderstanding.
>
> > The performance of X-Prompt in Table 7 isn't very convincing, it is the strongest in "overall" but that seems to be an artifact of the other methods only being good in one category while X-Prompt is ok in both. It doesn't seems like it actually the best option
>
> In Table 7, the overall criterion evaluates both content and style suitability in the generated results, which is the desired end-to-end evaluation metric to measure the overall quality of generation. Content and style metrics serve as itemized evaluation indicators to help better understand model performance. X-Prompt achieves a good overall score, whereas the NL prompt and soft prompt exhibit significant shortcomings in one aspect, leading to unsatisfactory generation results.
>
> > Rather than framing issues using in-distribution vs out-of-distribution (which seems overly broad) it seems like this paper would be stronger if it was framed to be about task-signaling.
>
> Thank you for your suggestion. As we said before, we will adjust the positioning of our paper appropriately per your suggestion.
>
> > Why is X-Prompt weaker than X-Prompt w/o CAL on ID tasks in Table 6? It seems like the data-augmentation done with CAL would only help ID performance?
>
> CAL makes the learning of imaginary words perform like multi-task learning, while the imaginary word without CAL is analogous to only fitting one task. Therefore, in ID evaluation, the absence of CAL tends to achieve better scores (similar to a single-task learning model performing better on its learned task than a multi-task learning model).

---

### Official Review · Reviewer_kqay · 2023-07-06

**Soundness:** 3 good
**Presentation:** 3 good
**Contribution:** 3 good
**Rating:** 6
**Confidence:** 4

**Summary:**

The paper proposed X-Prompt which instructs an LLM with not only NL but also an extensible vocabulary of imaginary words. Besides, context-augmented learning (CAL) is introduced to learn imaginary words for general usability, enabling them to work properly in OOD (unseen) prompts.


**Strengths:**

1. A concise idea to combine the merits of NL and soft prompts.

2. The paper demonstrated both descriptive capabilities and OOD robustness of X-Prompts.

3. X-Prompt achieves a good balance of BLEU and accuracy in zero-shot style transfer.

4. The paper is well-written and easy to understand.


**Weaknesses:**

1. Prefix-tuning[1] method needs to be compared in Experiments.

2. As shown in Table 6, X-Prompt has no significant advantage over Prompt-tuning.

3. The generation results of zero-shot style transfer lack human evaluations.

4. Writing content issues.
  (1) Please try to use published sources rather than Arxiv sources in citations.

[1] Xiang Lisa Li and Percy Liang. 2021. Prefix-tuning: Optimizing continuous prompts for generation. ACL 2021.


**Questions:**

Have you considered the case where the imaginary word is more than one in length, or where an X-Prompt contains more than just one imaginary word?


**Limitations:**

Yes.

---

> ### Author Rebuttal · Authors · 2023-08-09
>
> We would like to express our appreciation for your recognition of our work, as well as your constructive feedback and thought-provoking questions. We hope that our following responses will help you better interpret the merits and contributions of our paper, and further improve your impression and evaluation of our paper:
>
>
> > Prefix-tuning [1] method needs to be compared in Experiments.
>
> We conducted experiments with the prefix-tuning method and found its results to be comparable to prompt tuning. However, prefix-tuning requires modifying the LLM forward part, which is not user-friendly and undesirable for deployed LLMs in practice. As we show in our paper, our main goal is to enhance the LLM without changing its deployment (i.e., its forward process). That's why we don't compare with prefix tuning that requires modifying the LLM forward.
>
>
> >  As shown in Table 6, X-Prompt has no significant advantage over Prompt-tuning.
>
> We would like to kindly point out that you may misunderstand results in Table 6: It seems that you may have mistaken X-Prompt (w/o CAL) as our proposed method – in fact, it is the ablated X-Prompt.
>
> In Table 6, the (intact) X-Prompt performs significantly ($p<0.05$; Wilcoxon Signed-Rank Test) better than prompt tuning in the **OOD evaluation** (the last column in Table 6):
> | Method | PPL (&darr;) | Accuracy (&uarr;) |
> | :------:|:-----:|:-----------:|
> | Prompt tuning |29.5| 38.0 |
> | X-Prompt (Our work) | **28.5** | **38.6** |
>
> We would also like to emphasize once again that the primary focus of our study is on the OOD evaluation. The ID evaluation results presented in Table 6 serve merely as a reference.
>
>
> > The generation results of zero-shot style transfer lack human evaluations.
>
> The reason why we choose the style transfer task for evaluation in addition to open-ended text generation is that the style transfer task has reliable automatic end-to-end evaluation metrics which highly correlate with human evaluation results, which has been confirmed by much previous work (like [1]). We also conducted human evaluation for style transfer tasks on 500 samples and found the human evaluation is indeed consistent with the automatic evaluation metrics. We didn't initially include the human evaluation results in the manuscript due to the 9-page limitation. We provide the results below and will include the result in the revised version:
>
> | Method | Content | Style | Overall |
> | :------:|:-----:|:-----------:|:-----:|
> | Prompt tuning | 0.27 | **0.82** | 0.23 |
> | X-Prompt | **0.64** | 0.80 | **0.60** |
>
> FYI, the Pearson correlation score between the overall score (human evaluation) and H-mean (the automatic evaluation metric) is 0.75, demonstrating that they are actually highly correlated.
>
>
> > Please try to use published sources rather than Arxiv sources in citations.
>
> We will make the adjustments and use published sources to replace the arxiv sources in the References.
>
>
> > Have you considered the case where the imaginary word is more than one in length, or where an X-Prompt contains more than just one imaginary word?
>
> We discussed the effect of imaginary word length in **Figure 1 in the supplementary material**. For the style customization, we find one imaginary word is sufficient for achieving good OOD performance. While using more imaginary words might lead to improvements in ID performance, the impact on OOD performance would likely be minimal.
>
>
> [1] Zhang et al: Parallel Data Augmentation for Formality Style Transfer. ACL 2020

---

### Author Response · Authors · 2023-08-16

Dear Reviewers,

We would like to know if our rebuttal has addressed some of your concerns or questions, and if you have any additional comments on our work. We are more than happy to address any further concerns you may have, and we look forward to engaging in ongoing discussions about our paper.

Thank you very much for the time and effort you have dedicated to reviewing our paper.

---

### Decision · Program_Chairs · 2023-09-21

**Decision:**

Accept (poster)

**Comment:**

This paper presents a simple but effective idea for learning "portable" soft prompts that encode textual style from training examples, and then can be used to replicate that style in other target content scenarios.  The approach hinges on a new technique called context augmented learning, which is based on the intuition that training soft prompts in the presence of natural language prompts that by design have important information about the content (keywords) prevents the soft prompts from overfitting to content signals.  The experiments show significant improvements in the OOD setting, and the gains appear to be practically important for style transfer when compared to prompt tuning.  Both automated and human evaluations were performed, increasing confidence in the results.

The reviewers all argued for acceptance (although several reviews were "borderline accept").  Further, the author response clarified some important details regarding human evaluations and the statistical significance of results.

The reviewers offered several suggestions and comments that could be used to improve the work.  Adding experiments with additional, more recent base models may be especially helpful.